# Nanoparticle-Based Approaches in the Diagnosis and Treatment of Brain Tumors

**DOI:** 10.3390/jcm13237449

**Published:** 2024-12-06

**Authors:** Parvin Pourmasoumi, Seyed Abdolvahab Banihashemian, Farshid Zamani, Aghdass Rasouli-Nia, Davood Mehrabani, Feridoun Karimi-Busheri

**Affiliations:** 1Department of Biomedical Engineering, Central Tehran Branch, Islamic Azad University, Tehran 19395-1495, Iran; pari.pourmasoumi@gmail.com (P.P.); vssabani@gmail.com (S.A.B.); 2Stem Cells Research Center, Tissue Engineering and Regenerative Medicine Institute, Central Tehran Branch, Islamic Azad University, Tehran 14778-93780, Iran; 3Department of Immunology, School of Medicine, Shahid Beheshti University of Medical Sciences, Tehran 19839-69411, Iran; farshidzamani70.scu@gmail.com; 4Department of Oncology, Faculty of Medicine, University of Alberta, Edmonton, AB T6G 1Z2, Canada; arniapnk17@gmail.com; 5Stem Cell Technology Research Center, Shiraz University of Medical Sciences, Shiraz 71348-14336, Iran; 6Burn and Wound Healing Research Center, Shiraz University of Medical Sciences, Shiraz 71348-14336, Iran; 7Comparative and Experimental Medicine Center, Shiraz University of Medical Sciences, Shiraz 71348-14336, Iran

**Keywords:** brain tumor, neuro-oncology, *Glioblastoma multiforme*, nanoparticle, treatment

## Abstract

Glioblastomas are highly invasive brain tumors among perilous diseases. They are characterized by their fast proliferation and delayed detection that render them a significant focal point for medical research endeavors within the realm of cancer. Among glioblastomas, *Glioblastoma multiforme* (GBM) is the most aggressive and prevalent malignant brain tumor. For this, nanomaterials such as metallic and lipid nanoparticles and quantum dots have been acknowledged as efficient carriers. These nano-materials traverse the blood–brain barrier (BBB) and integrate and reach the necessary regions for neuro-oncology imaging and treatment purposes. This paper provides a thorough analysis on nanoparticles used in the diagnosis and treatment of brain tumors, especially for GBM.

## 1. Introduction

Brain tumors are among the cancers with special research interest that pose extensive challenges in both the diagnosis and treatment of patients. Brain tumors can lead to fatal complications, especially in advanced stages [1]. In 2016, the National Brain Tumor Society demonstrated that around 78,000 individuals in the United States received services for the diagnosis of brain tumors, while 16,616 fatalities were noticed among these patients [2]. Among them, primary brain tumors arose from different types of cells within the central nervous system (CNS). Distinctive genetic polymorphisms, exposure to ionizing radiations, and chemical carcinogens were illustrated as key factors in the development of brain tumors [3,4,5]. Glioma is a primary brain tumor that is classified according to the cells of origin including astrocytic tumors, oligodendrogliomas, ependymomas, and mixed gliomas. Among astrocytic tumors, glioblastoma is one of the rarest gliomas with high mortality that arises from the supportive cells, most commonly astrocytes that surround and protect neurons in the brain [4,6].

Despite recent advances in imaging and treatment methods, the prognosis for many patients with brain tumors still remains dismal and necessitates innovative strategies to improve outcomes [5,6]. Grade 4 adult gliomas are IDH-mutant astrocytomas and IDH-wildtype glioblastomas having a very high mortality rate, while survival at 5 years does not exceed 5%. Patients’ comorbidities such as hypertension, diabetes mellitus/hyperglycemia, a subtotal surgical resection, degree of cell differentiation, and CDKN2A gene mutations were described as negative prognostic factors that give them such aggressiveness [7].

In recent years, nanoparticles have gained attention in the management of brain tumors. They have unique physicochemical properties that have revolutionized the landscape of neuro-oncology including imaging [8]. Nanoparticles have a small size, specific surface chemistry, and adjustable optical properties that, in imaging probes, increases their accuracy and specificity in the diagnosis of brain tumors [9]. The technologies of magnetic resonance imaging (MRI), positron emission tomography (PET), and fluorescence imaging were shown as common methods in the detection of nanoparticles [10]. Nanoparticles are versatile platforms for the non-invasive imaging of brain tumors to help facilitate early detection characteristics, accurate localization, and timely monitoring in tumor proliferation [8,9].

Traditional chemotherapy methods for brain tumors often have limitations in delivering therapeutic agents to the tumor site due to the impermeable nature of the blood–brain barrier (BBB), and they may also carry an increased risk of systemic toxicity. So, drug delivery systems based on nanoparticles were demonstrated to solve this problem and provide the possibility of the encapsulation, protection, and transport of therapeutic agents through the brain towards the tumor cells. Though the exact mechanism of the transport of therapeutic agents into the brain is not understood, it was found to be dependent on the particle size, structure, material composition, and design of nanoparticles [11]. When nanoparticles are targeted with ligands such as antibodies or peptides, the process of delivering drugs to specific locations is facilitated and leads to an increased therapeutic efficiency, reduced side effects, and significant improvements in the management of brain tumors. In this relation, chemotherapy medications could effectively diminish the tumor size and edema together with the elimination of cancer cells (Figure 1). Radiation therapy has also been employed to circumvent brain malignancies by utilizing accurate and concentrated radiation to specifically target the tumor area while safely preserving the neighboring brain tissues [9,12].

The combination of procarbazine, temozolomide, lomustine, and vincristine has successfully been utilized in radiation therapy [13]. Nevertheless, the BBB still hinders the effective transportation of drugs to the CNS [14], while targeting anticancer drugs still presents challenges that must be addressed such as the biocompatibility, pharmacokinetics, and long-term safety of nanoparticle formulations [15]. Furthermore, the translation of nanoparticle-based formulations from preclinical trials to clinical practice still requires rigorous verified clinical trials [16,17]. Nanocapsules employing clustered regularly interspaced short palindromic repeats (CRISPR)-associated (Cas) nuclease 9 (CRISPR-Cas9) have high efficiency in delivering drugs to the brain and in the treatment of cancer tumors [18]. CRISPR/Cas9 is a precise and remarkable site-specific gene editing tool with a breakthrough in the field of biotechnology. It possesses three distinct forms of deliverable CRISPR, viz., plasmid DNA (having Cas9 and sgRNA insert), messenger RNA (mRNA) (Cas9 expressing), and the main form, i.e., ribonucleoprotein (RNP) complexes that enroll a Cas9 effector along with target-specific sgRNA; they all may have shortcomings too [19].

The nanocapsules have nearly neutral surface charges to protect the CRISPR-Cas9 from RNase and prolong the lifetime of the nanoparticles in blood circulation. Also, the drug-loading efficiency is very high, almost 100%; thereby, they can enhance the delivery efficacy. In addition, nanocapsules have a small size (around 30 nm) and the rational modification of angiopep-2 peptide as a peptide with good affinity with lipoprotein receptor-related protein-1 (LRP-1). LRP-1 is highly expressed on BBB endothelial cells and glioblastoma cells that allow nanocapsules to cross the BBB and to efficiently undertake the intracellular delivery to GBM cells [20]. Moreover, the disulfide bond ensures that the nanoparticles can selectively degrade inside the tumor cells via the extraordinarily high glutathione (GSH) concentration, which results in the precise release of drugs (Figure 2) [20].

The modification of GBM biology via the gene editing of CRISPR/Cas9 has resulted in successful outcomes. Fierro et al. successfully used a knockout strategy to target PD-L1 that led to a 64% decrease in relation to the PD-L1 protein levels of U87 cells [21]. Lumibao et al. employed a CRISPR-Cas9 knockout of Coiled-Coil-Helix-Coiled-Coil-Helix Domain with 2 (CHCHD2) in epidermal growth factor receptor-vIII (EGFRvIII)-expressing U87 cells and showed changes in mitochondrial respiration and glutathione condition and a decline in cell growth and invasion [22]. Toledano et al. noticed positive findings via a Plexin-A2 knockout demonstrating the Plexin-A2′s pro-proliferative impacts to be mediated by FERM domain, ARH/RhoGEF and Pleckstrin Domain Protein 2 (FARP2), FYN as a tyrosine-specific phospho-transferase, and the guanosine triphosphate enzyme (GTPase) activating in the intracellular domain [23]. Also, Guda et al. targeted RGS4 through a knockout method and reported a reduction in cell proliferation [24]. Zhang, in a knockdown approach on Nanos3, showed the decreased in vitro proliferation, migration, and invasion of GBM cells, an in vivo increase in sensitivity to doxorubicin (Dox) and temozolomide (TMZ), and the inhibition of tumor growth [25]. In the field of the CRISPR/Cas9 gene editing of GBM, successful outcomes in modulating angiogenesis were illustrated [26]. Raghu et al., in a knockdown approach, targeted and down-regulated Notch1 with a significant increase in comparison to control xenografts [27]. When Podoplanin (PDPN) was targeted by a knockout approach, identical proliferation rate, apoptosis, angiogenesis, and invasion were noticed in Podoplanin-deleted tumors in comparison to the controls [28]. Szymura et al., when working on DExD/H-Box Polypeptide 39B (DDX39B) as a knockdown strategy, demonstrated that the depletion of CRISPR-mediated DDX39B could result in an increase in p65 phosphorylation and could render U87 cells that were very resistant to TMZ [29]. Lu et al. used the knockdown of Brefeldin A (BFA)-inhibited guanine nucleotide-exchange protein (BIG)1 and 2 to significantly reduce the expression of vascular endothelial growth factor (VEGF) mRNA and protein levels in GBM U251 cells and human umbilical vein endothelial cells (HUVECs) [30]. Lee et al. targeted ANGPT2 using a knockout strategy and therapy utilizing agonistic anti-Tie2 antibody and 4E2, and exhibited the normalization of vascular tissue in GBM [31].

Therefore, there is a need for future research to improve nanoparticle designs and to promote operational methods for the integration of nanoparticles by employing technologies such as immunotherapy and gene therapy to verify the potential of nanoparticles in the management of brain tumors [16,17,19,20,21]. The main goal of this review was to provide comprehensive data on the use of nanoparticles in the diagnosis and treatment of brain tumors in order to illuminate the transformative potential of nanoparticles in neuro-oncology and in the management of brain tumors. This review can provide a deeper understanding for the role of nanoparticles in the accurate diagnosis of brain tumors, to target drug delivery, and, ultimately, to increase the therapeutic outcome in order to combat challenges in the treatment of brain tumors.

## 2. Nanotechnology in Brain Cancer Therapy

The development of new diagnostic and treatment approaches on the nano scale and technological advances in recent decades have had a great impact on the nano-oncology field. Inorganic and organic nanomaterials were shown to improve bioimaging techniques, targeted nanoparticle drug delivery systems (DDSs), contrasting agents, and diagnostic tools for the highly specific detection of macromolecules. Regarding their favorable physicochemical characteristics, many factors were mentioned in relation to the development of new diagnostic and treatment approaches to brain cancers in recent decade. They were described in terms of their small size, their large surface area relative to volume, their specific structural characteristics, the possibility of attaching different molecules to their surface, and their transformation into excellent transport vehicles to cross cellular barriers including the BBB [32,33].

Nanoparticle DDSs have gained significant interest for new, more effective, and less invasive methods in cancer therapy because of their potential to decrease the common side effects of conventional chemotherapeutic agents including the absence of any specificity and a premature drug release. Nanoparticle DDSs enroll the use of nanocarriers consisting of non-toxic monomers and polymers that display high physical and chemical stability and biocompatibility too. These nanocarriers can be altered to specifically target receptors of tumor cells and provide the potential for an accurate and site-specific drug delivery. Also, nanoparticle DDSs can be prepared to respond to many stimuli such as light, heat, temperature, pH, and ultrasound. These nanoparticles demonstrate biodegradability, biocompatibility, and a low toxicity. However, there are exceptions to biodegradability, like certain metal/metal oxide nanoparticles, whose biodegradability is dependent on coating and the synthesis methods employed. Many nanoparticle DDSs were extensively investigated such as exosomes, liposomes, polymeric nanoparticles, metal-organic frameworks (MOFs), quantum dots (QDs), dendrimers, and hydrogels. These nanocarriers present various structures and characteristics that suggest their unique advantages for drug delivery approaches [34].

Among nanocarriers, nanomedicines are small-sized nanocarriers that have been adopted to cure brain illnesses, including brain cancer and Alzheimer’s disease. Nanomedicines can easily interact with the proteins and molecules on the cell surface and inside the cell too. These nanomedicines are nanoparticle-functionalized ones that have central core structures to ensure the encapsulation or conjugation of drugs and to provide protection and a prolonged circulation in the blood stream. Nanomedicines are also specialized to target cells and an intracellular compartment to directly deliver the drugs at a predetermined dosage to the pathological site. Nanomedicines can cross the BBB and deliver the medicine to the expected regions (Figure 2) [35].

Due to recent developments in materials sciences and nanotechnology, various strategies were developed in the BBB as well as a library of brain-targeted drug delivery systems (Figure 3). The transport routes of the drug molecules across the BBB occur via many pathways, including paracellular and transcellular diffusion, receptor-mediated transcytosis, cell-mediated transcytosis, transporter-mediated transcytosis, and adsorptive-mediated transcytosis (Figure 3) [36]. The BBB is considered a neurovascular unit composed of vascular endothelial cells with surface charge modifications, tight junction proteins, pericytes, astrocytes, and other components that are selectively permeable and can block solutes in the systemic blood to enter the CNS [35].

### 2.1. Liposomes

Liposomes, one of the most studied nanomaterials, are nano-scale spheres composed of either synthetic or natural phospholipid bilayers with aqueous cores. Due to the amphiphilic nature of phospholipids, liposomes are formed spontaneously and they are synthesized by thin-film hydration or reverse-phase evaporation [39]. Liposomes are either unilamellar (as small as 100 nm and as large as 200–800 nm) or multilamellar (500–5000 nm, consisting of many lipid bilayers with the same center) [32]. According to the liposome’s structures, they are classified into four categories based on size and number of bilayers including small unilamellar vesicles (SUV), large unilamellar vesicles (LUV), multilamellar vesicle (MLV), and multivesicular vesicles (MVVs). Liposomes have a mono phospholipid bilayer in a unilamellar structure, while the multilamellar structure has an onion-like structure.

MVVs form a multilamellar arrangement with concentric phospholipid spheres as many unilamellar vesicles are produced within larger liposomes. The efficiency of liposome encapsulation increases with liposomal size, and decreases with the number of bilayers of hydrophilic compounds. The size of the vesicles is an important factor that controls the half-life circulation of liposomes, while both the size and number of bilayers can influence the amount of the encapsulated drug. When liposomes are employed for drug delivery, the desired vesicles usually extend from 50 nm to 150 nm and their interaction with the cell membrane is represented by various pathways including specific (modified with receptor-mediated) or nonspecific endocytosis, local fusion (adhesion), phagocytosis, and absorption into the cell membrane. Liposome–cell interactions are influenced by a variety of factors such as composition, the diameters of liposomes, surface charge, targeting ligands on the liposome surface, and biological environment [40].

The delivery of liposomes to the brain is undertaken through different ways such as intranasal, intracarotid, intracranial, and intraperitoneal injections and the method of convection-enhanced delivery (CED). As the carotid artery is considered a major blood vessel to carry blood from the heart to the brain tissue, the intracarotid injection method can lead to the direct injection of the drug into the carotid artery; the intranasal injection route can bypass the BBB to deliver the drug via the nose to the brain tissue; the intracranial injection approach can be an effective way for the direct delivery of drugs to specific regions of the brain tissue; and the intraperitoneal injection method can provide delivery of the drug via the peritoneal route. The CED can also establish a pressure gradient on the tip of the infusion catheter implanted in the brain tissue, administering drugs directly into the interstitial spaces of the brain tissue [34]. Liposomes were shown to have the capacity to improve drug delivery to brain tumors and to enhance the treatment outcome. They represent their therapeutic influence via the release of their cargo in specific areas of the tumor’s vasculature and extravascular space (Figure 4). Using a targeted delivery, liposomes can be directed to specific regions of the BBB or GBM tumors and to deliver anti-cancer drugs.

Table 1 discusses studies aimed at translating these promising formulations into preclinical settings [36]. Dox, TMZ, CB, CPPs, Tf, DTX, QDs, and CPT-11 have been utilized as drug payloads, and receptor-mediated endocytosis and TML as mechanisms of entry with high efficacy. It was shown that these drugs, except for CPPs and Tf, could improve survival in vivo. For drug delivery to the brain, non-targeted liposomes have also been investigated. Liposomal DOX (Myocet) has been used in the treatment of recurrent gliomas in children (NCT02861222), and liposomal irinotecan in the treatment of recurrent high-grade GBM (NCT02022644) [41]. In 2008, a Phase I study evaluated the therapeutic potential of liposomal irinotecan (NL CPT-11) and reported the failure of the therapeutic potential of liposomal irinotecan for Phase II trial in GBM patients [42]. When another liposomal irinotecan (MM-398) was utilized, a significant anti-tumor effect was demonstrated in patients suffering from advanced breast cancer with metastases to the brain (NCT01770353) (Table 2) [43]. It seems that liposomes have the capacity to improve drug delivery to brain tumors and to enhance therapeutic outcome. They represent their treatment efficacy via the release of their cargo through a targeted delivery. So, liposomes can be directed to specific regions of the BBB or GBM tumors to deliver anti-cancer medications.

Liposomes loaded with erlotinib and doxorubicin showed promising effects in the treatment of brain tumors [51]. These liposomes, when modified with transferrin, demonstrated significant accumulation in the brain and resulted in tumor regression in mice models. Liposomes coated with a vitamin E derivative of tocopheryl polyethylene glycol succinate (TPGS) were also developed for docetaxel delivery with a better performance than the traditional liposomes. These liposomes are a promising drug delivery system for brain tumors [52]. Immunoliposomes, which are liposomes modified with antibodies, have also held significant promise in cancer therapy. The Food and Drug Administration (FDA) has approved the monoclonal antibodies (mAbs) that have been designated for cancer treatment [53]. In this relation, particularly in brain tumors, where vascular endothelial growth factor (VEGF) and its receptor VEGFR2 are overexpressed, immunoliposomes can offer a targeted approach to combat angiogenesis and metastasis [54].

Paclitaxel, with the efficacy in targeting microtubules and combating a range of cancers such as lung and ovarian, and brain tumors, has been demonstrated to have limitations in the treatment of glioma due to its inadequate penetration of the BBB [55]. The release of paclitaxel from microspheres has been tailored by controlling many parameters such as polymer type, fabrication technique, drug loading content, stabilizer concentration, and the use of additives to control porosity. While paclitaxel has been used to treat other cancer types, the systemic delivery of this drug for the treatment of glioma has failed because of the low levels of paclitaxel that reach the gliomas and due to the poor penetration of the BBB [56,57]. The effect of the concentration of the stabilizer on drug release was studied by Elkharraz et al. for paclitaxel-encapsulated poly(lactic-co-glycolic acid (PLGA) microspheres. An increase in polyvinyl acetate (PVA) concentration was shown to decrease the sphere diameter due to the reduction in surface tension between the organic and aqueous phases and an increase in external aqueous viscosity. With the smaller spheres having a faster release, they can affect the drug release due to the decreased lengths of diffusion, regardless of the drug loading [58].

To address this challenge, researchers have developed nanotherapeutic systems to traverse the BBB, to target vascular mimicry channels, and to eradicate brain tissue stem cells. A novel liposomal formulation incorporating paclitaxel alongside the artemether was shown to have anti-tumor and apoptotic regulatory properties and to represent a promising advance in this endeavor [55]. So, based on liposome characteristics, they are also applied in other approaches such as gene delivery by packing DNA in liposomes that contain cationic lipids (e.g., Dioleoylphosphatidylethanolamine (DOPE) or N-[1-(2,3-dioleyloxy)propyl]-N,N,N-trimethylammonium chloride (DOTMA)). They are also an excellent imaging tool. In computerized tomography (CT), they are employed to carry a contrasting agent of iodine to detect hematological diseases and tumors. Similarly, gadolinium can be packed in liposome to increase half-time and contrast during MRI. Liposomes can also be formulated to contain air bubbles and to be applied in ultrasound imaging [32].

### 2.2. Dendrimers

Dendrimers, as hyper-branched nano-sized polymers, are recognized for their organized three-dimensional structure and extensive surface area, making them promising drug carriers. These polymers offer advantages such as nanoscopic size, stability, rapid cell entry, and targeted drug delivery. Functional groups on dendrimer surfaces allow for the effective attachment of therapeutic molecules. Surface modifications like glycosylation and pegylation can improve their biocompatibility and safety. Therefore, targeted ligands can be added to dendrimer surfaces for selective delivery to brain tumor cells, sparing normal cells. Designing new dendrimer types and enhancing the drug delivery capabilities and improvement of the biocompatibility for broader therapeutic applications are current topics of interest [59,60,61]. Dendrimers, which possess unique properties such as nanosized defined composition and programmable surface functions have been extensively investigated for brain tumor therapy as poly(amidoamine) (PAMAM), polypropylenimine (PPI), poly-l-lysine (PLL), carbosilane, phosphorus, peptide, glycodendrimers, triazine, polyglycerol, citric acid, polyether, and surface-tailored dendrimers [56].

Dendritic polymers, specifically PAMAM dendrimers, have become highly significant in the field of cancer therapy because of their strong and precise nano-polymeric structures, which enable the targeted delivery of genes and drugs. Consequently, in the past several years, numerous nano theragnostic drug delivery systems utilizing PAMAM dendrimers have been created to harness the potential for precise drug administration in brain treatment [57]. Sharma et al. devised a novel strategy to effectively deliver TMZ to the brain in the treatment of glioblastoma. They utilized PAMAM dendrimers to formulate a PAMAM-chitosan conjugate-based nanoformulation of TMZ called PCT. Testing this formulation in vitro and in vivo revealed that this formulation was more effective against glioma cell lines U-251 and T-98G than the pure TMZ alone. Characterization via 1H Nuclear Magnetic Resonance (NMR), Fourier Transform Infrared (FT-IR) spectroscopy, and surface morphology assessment confirmed its suitability, while the safety was ensured in an in vivo evaluation. Pharmacokinetic analysis demonstrated sustained release properties, with PCT showing a longer half-life in comparison to TMZ alone. A bio-distribution study revealed a higher concentration in the heart than the brain. Overall, the study suggested that utilizing chitosan-anchored dendrimers for TMZ delivery could hold promise for enhancing glioblastoma therapy [62]. With the help of dendrimers, nucleic acids and drugs can be sent to the brain and cancer cells without using any viruses due to the highly branched structure and the available internal cavities of these polymers, which make them an excellent delivery system for genes and drugs [56].

## 3. Nano-Micelles

Several nanomedicines are utilized for the delivery of therapeutic agents in the treatment of brain cancer such as (i) nanoparticles, (ii) niosomes, (iii) liposomes, (iv) dendrimers, (v) micelles, etc. Among all nanomedicines, micelles are one of the best suitable nanocarriers. They facilitate the higher penetration of therapeutic drugs of the BBB, reduce multidrug resistance, and inhibit tumor recurrence post-surgery [63]. It can be prepared from amphiphilic di/tri-block or from graft copolymers and offers a high pay-load and greater biocompatibility both in vitro and in vivo. Micelles are highly capable of loading and administering hydrophilic and hydrophobic drugs to the body organs and can help increase the pharmacokinetic and bioavailability profiles [6]. Micelles are absorbed through passive diffusion and the receptor-based endocytosis mechanism. This has clearly illustrated its Enhanced Permeability and Retention effect (EPR) into tumor tissues [63].

Nano-micelles represent a promising class of amphiphilic nanostructures formed through the self-assembly of amphiphilic molecules in aqueous environments above a critical micelle concentration. These structures possess both hydrophilic and hydrophobic regions, with hydrophilic molecules forming the shell and hydrophobic regions creating the core, where lipophilic substances can be encapsulated. Nano-micelles show great potential as carriers for chemotherapeutic agents in targeted ovarian cancer therapy to present significant penetration and endocytosis properties in cancer cells while minimizing the non-specific targeting of normal cells [64,65]. They can boast stability, improve biological compatibility, prolong plasma circulation, and enhance penetration into inflammatory and tumor tissues.

In recent years, micelles have emerged as promising vehicles for the treatment of brain cancer due to their nano-scale dimensions, which enable them to provide clearance by phagocytosis and to penetrate the BBB [66]. Gao et al. developed a novel approach for glioma therapy using methoxy poly (ethylene glycol)-poly(ε-caprolactone) (MPEG-PCL) nanoparticles co-loaded with honokiol (HK) and Dox. The nanoparticles of 34 nm in size could efficiently deliver both drugs, revealing an in vitro extended release profile. HK-Dox-MPEG-PCL micelles could effectively suppress the proliferation of glioma cells and induce apoptosis. In animal models, these micelles outperformed single-drug formulations, inhibited glioma growth, and enhanced anti-tumor properties, suggesting the promising clinical applications of HK-Dox-MPEG-PCL micelles in glioma therapy [66].

Vitamin E D-α-tocopheryl polyethylene glycol succinate (TPGS) micelles loaded with docetaxel and transferrin have been formulated for brain tumor therapy, as transferrin-conjugated micelles exhibited a high drug loading capacity and remarkable kinetics for drug release that can result in increased and improved drug delivery to the brain when compared to non-targeted formulations [67]. Huang and his team devised a potent treatment strategy for GBM as a challenging brain tumor by developing amphiphilic CB-poly ethylene glycol (PEG)-Ce6 polymer to form Cetylpyridinium chloride (CPC) micelles loaded with 5-(3-methyltriazen-1yl)-imidazole-4-carboxamide (MTIC) as a chemotherapy drug. Coating these micelles with hybrid membrane mUMH as a hybrid membrane (HMC3 membrane: macrophage membrane: U87MG membrane = 1:1:2) could enhance the targeted delivery of GBM. MTIC release is triggered by GBM microenvironment cues and can facilitate an effective chemotherapy. In a mouse model, mUMH@CPC@MTIC demonstrated superior anti-tumor efficacy, suggesting the promising prospects of GBM therapy [67].

## 4. Carbon Nanotubes (CNTs)

CNTs are cylindrical nanostructures that have attracted the attention of pharmaceutical researchers based on their unique mechanical, electrical, and surface properties. These characteristics render them favorable candidates for easing the administration of medicinal chemicals and drugs. The surface characteristics of CNTs enable easy modification with different chemicals, thus enhancing their compatibility with biological systems. The functionalized nanocarriers can undergo modifications with polymers, carbohydrates, peptides, and organic chemicals, offering the potential for utilization in cancer treatment and targeted therapy against tumor cells [68,69]. Ren et al. developed a dual-targeting drug delivery system using PEGylated oxidized multi-walled carbon nanotubes (O-MWNTs) modified with angiopep-2 (ANG-2) as O-MWNTs-PEG-ANG for the treatment of brain glioma. O-MWNTs-PEG-ANG could effectively target glioma cells and cross the BBB. These nanocarriers, when loaded with Dox, exhibited enhanced anti-glioma effects compared to Dox alone and with low toxicity [70]. Han et al. conducted a study to evaluate the potential toxicity and underlying mechanisms of multiwalled carbon nanotubes (MWCNTs) on C6 mouse glioma cells. The exposure of C6 cells to MWCNTs resulted in a concentration- and time-dependent decrease in cell viability, the induction of apoptosis, G1 cell cycle arrest, and increased levels of oxidative stress, especially at higher concentrations, which suggests that smaller MWCNTs exhibiting higher toxicity are potentially mediated by increased oxidative stress [71].

## 5. Silver Nanoparticles (AgNPs) and Gold Nanoparticles (AuNPs)

AgNPs and AuNPs have garnered attention for their unique properties in nanotheranostics applications. These nanoparticles offer biocompatibility, shape versatility, integration with functional groups, and potential in cancer therapy and diagnosis. Alongside their benefits, AgNPs also exhibit cellular toxicity through oxidative stress. They boast properties such as monodispersity, tunable core size, facile synthesis, low toxicity, surface plasmon resonance absorption, large surface area, biomolecule binding ability, and the diagnostic potential. Therapeutic molecules can be loaded onto AgNPs via covalent bonds or electrostatic interactions, leveraging their multifunctional nature for efficient circulation and delivery to tumor cells or tumor surfaces [72,73,74]. Urbańska et al. investigated the potential of AgNPs in treating neuroepithelial tumors, particularly GBM, by using human GBM cells of the U-87 line cultivated on the chorioallantoic membrane of chicken embryos and treated with colloidal AgNPs at a concentration of 40 μg/mL. The results showed the hindered growth of GBM cells and with proapoptotic characteristics [73].

The caspase-9 monomer that consists of one large and one small subunit with catalytic domains, when dimerized as caspase-9 with the active site motif QACRG, one site closely resembles the catalytic side, whereas the second has no ‘activation loop’ that leads to the disruption of the catalytic machinery in that particular active site [11,13]. Noticeable variations in the rates of cell death and the levels of active caspase 9 and active caspase 3 were reported between the group treated with AgNPs and the control group. The inhibitory effect of AgNPs on cell growth has also been more prominent than their impact on cell death [72]. Yu et al. developed TMZ-loaded gold nanoparticles (TMZ@GNPs) modified with Anti-Erythropoietin-Producing Hepatocellular Carcinoma A3 (EphA3) to combat TMZ resistance in glioblastoma. These nanoparticles increased in vitro cellular uptake and induced apoptosis in T98G cells when subjected to laser irradiation. An in vivo study on nude mice with GBM models showed prolonged survival with anti-EphA3-TMZ@GNPs when compared to TMZ alone. This approach holds promise for overcoming TMZ resistance and improving GBM treatment [75].

## 6. Advances in Brain Tumor Diagnosis

### Nanomaterials for Brain Cancer Diagnosis and Biosensing

Nanotechnology offers promising avenues for the detection and biosensing of several cancers. In this relation, nanoparticles, with their unique properties, serve as ideal contrast agents for the precise imaging of many tissues in vitro and in vivo [10,76,77]. Different nanomaterials have been introduced to improve their dispersion, accumulation in certain tissues, and elimination from the body. As nano-diagnostics, these nanoparticles can undergo phagocytosis by brain tumor cells to reach the high-resolution imaging of malignant tissues, and they can be effectively differentiated from healthy cells too. Novel methods were demonstrated to precisely mark/pinpoint tumor tissue via controlling the targeting of nanostructures and by utilizing specific coatings [32]. Optical, photoacoustic, and fluorescence imaging methods, positron emission tomography, computed tomography, and MRI were described as conventional approaches to possess both benefits and constraints. Neurophotonics technology such as Raman spectroscopy and fluorescence spectroscopy hold the potential to improve the detection and diagnosis of brain tumors (Table 2).

## 7. Nanomagnetic Particles

Nanomagnetic particles (N-MNPs) offer distinct properties that make them reliable tools for cancer detection. Among them, superparamagnetic iron oxide nanoparticles (SPIONs) and ultra-small superparamagnetic iron oxide nanoparticles (USPIONs) were shown to have stability, magnetic sensitivity, and unique physical characteristics [10,33,76,77]. These features render them suitable for various applications, including intracellular cancer imaging and tumor detection [94]. It was demonstrated that folic acid (FA)-coated SPIONs can be conjugated with bovine serum albumin (BSA) for MRI and the detection of glioblastoma. The biocompatibility and cellular internalization characteristics of FA-BSA-SPIONs can exhibit outstanding outcomes in U251 glioma cells. Nevertheless, nanomaterials offer a prominent platform for the precise imaging of brain tumors or cancer [95].

An analogous study employed polymer hybrid magnetic nanoparticles (MNPs) that were labeled with a fluorescent dye to investigate and pinpoint human GBM. SPIONs that were modified with poly(lactic-co-glycolic) acid (PLGA) and labeled with fluorescein isothiocyanate (FITC)-labeled polyethyleneimine (PEI) have been utilized too. Human GBM U251 cells exhibited a stronger uptake of labeled MNPs in comparison to unlabeled ones. These MNPs functioned as a highly efficient means for seeing cells and delivering drugs at the same time, without causing any harm to healthy cells [96]. A noteworthy discovery was the utilization of a versatile nanoprobe consisting of PEGylated USPIONs that were linked with angiopep-2 to specifically target the microvasculature of glioblastoma. Angiopep-2 exhibited a strong attraction to the low-density lipoprotein receptor-related protein that was excessively produced in glioblastoma. The nanoprobes exhibited a high level of biocompatibility and successfully visualized intracerebral glioblastomas by penetrating BBB. This holds great potential for a non-invasive diagnostic tool in the future. These observations describe the unique properties and applications of N-MNPs in cancer imaging without any similarity to other particles [97].

## 8. Extracellular Vesicles (EVs) and Exosomes

Glioma, a type of brain cancer, requires dependable non-invasive methods for diagnosis and monitoring. EVs are body small subset cells that release nano-scale particles and are surrounded by a membrane with their vital function in the transportation of molecular constituents from cells that can facilitate their recognition and detection. They contain DNA, RNA, lipids, and proteins that can cross the BBB by means of transcytosis [98,99]. In glioma, EVs containing EGFR proteins can offer a specific marker for the tumor and the precise detection of the tumor [100]. Biomimetic nanostructures coated with lipids similar to those on EVs show an efficient therapeutic approach to selectively target malignant cells [101].

Also, exosomes, which are lipid bilayer extracellular vesicles, are released from different cell types and have a nano-scale structure. They have both coding and non-coding RNAs as well as lipids, which make them a powerful and safe diagnostic tool to cross the BBB [102]. A recent study investigated the utilization of SPIONs loaded into specific exosomes derived from glioma cells to enhance and facilitate imaging modalities. The study targeted the neuropeptide-1 associated with the exosome membrane. Additionally, curcumin was encapsulated within exosomes to leverage its medicinal properties. Magnetic flow facilitated the accumulation of SPIONs in tumor tissues to enable their simultaneous use in imaging and for therapeutic purposes [103].

## 9. Multimodal Imaging with Nanomaterials

Limited accessibility to the BBB is a hurdle that can prevent the successful identification and treatment of brain tumors. Metastases often occur in advanced stages of cancer and require difficult surgical interventions; therefore, early detection is critical for optimal therapeutic approaches. The efficacy of nanodiagnostics is dependent on a well-established imaging technique that can accurately assess the pharmacokinetic profile, bioavailability, tumor neovascularization, uptake by tumor cells, kinetics of drugs, and release of imaging agents [56,104]. A wide range of non-invasive imaging modalities known as molecular imaging have been utilized to visualize, interpret, and assess the physiological changes at the molecular/cellular/tissue level to gain insight into the mechanisms of oncogenesis [105]. Therefore, molecular imaging techniques were shown to be useful for assessing therapy response, to measure biodistribution, and to determine the drug release profile [106].

Several researchers have utilized nanomaterials in multi-modal imaging to enhance the precision and efficacy of diagnostic imaging in brain gliomas [107,108,109], as nanomaterials offer a high degree of accuracy in diagnostic and therapeutic purposes. Recently, dual or multi-modal imaging has been described as a successful approach to augment imaging sensitivity towards specific tissues by incorporating ligands, markers, or aptamers. Moreover, these multi-modal techniques may also integrate external physical targeting technologies such as photoacoustic (PA) and MRI [110].

Meanwhile, gadolinium (Gd)-chelated diagnostic agents are extensively utilized as contrast agents in tumor imaging [111]. Currently, there are seven Gd contrast agents approved by the US-FDA for clinical use [112]. Ferromagnetic Gd contrast agents coupled with dendrimers have been utilized for image intensification and targeting while MRI monitoring is performed [113]. Dendrimers have been described in vivo as nanomedicines with the ability to tolerate bigger Gd charges and to improve the signal contrast of an MRI contrast agent [114]. Recently, Rasouli et al. investigated the potential of 99m Technetium-labeled dendrimer-phenylalanine conjugates in C6 glioma cell lines for the diagnosis of brain tumors using single photon emission computed tomography (SPECT). They showed that these dendrimers do not exhibit any toxicity in the brain, whereas phenylalanine increased the accumulation and deposition of 99m Technetium-labelled dendrimer in brain tumors [115] (Table 3).

Mathiyazhakan et al. deliberately engineered a liposome carrier possessing theranostic characteristics to selectively target glioma tissue via magnetic guiding. This vehicle utilizes QDs and SPIONs in combination with a silencing peptide to inhibit the formation of new blood vessels in tumor cells. The process of enclosing SPIONs and QDs into liposomes was successfully shown through the use of photon spectroscopy. The utilization of magnetic targeting resulted in an increased cellular uptake of these adaptable nanocarriers and led to a higher precision in the retrieval of glioma during surgical procedures [126]. In another study, the authors examined the impact of nanomaterials on brain tumor imaging by employing nanostructures containing near-infrared chemicals and USPIONs in order to conduct simultaneous multi-modal imaging and photothermal treatment. It was shown that the nanostructures could improve the efficacy of fluorescence imaging, as well as photoacoustic and magnetic resonance signals. In the study, a new nanostructure with enhanced photothermal contrast was created for multi-modal imaging in mouse models [127].

The benefits of magneto-fluorescent systems as a prospective diagnostic tool for serious brain cancers have been described before. Fluorescent magnetic nanotubes were used to create dual-modal probes through conjugation. Nanotubes that are sensitive to pH were demonstrated to enhance MRI capabilities by enabling a simpler viewing method for the spatial distribution of acidity. Animal experiments demonstrated a successful penetration of the BBB by multi-modal nanocomposites, which could efficiently enable the visualization of brain tumors and cancers by the utilization of MRI and fluorescence imaging [128]. Peptide-conjugated SPIONs have been utilized to visualize GBM by MRI and by optical imaging. The PEPHC1 peptide is linked to SPIONs to selectively bind to the EGFRvIII protein in GBM. These nanostructures exhibited that inherent biodegradability are non-toxic to living organisms and can be customized to improve tumor imaging [129]. Precise distinction between normal cells and gliomas is still a challenge. Therefore, integrating dual imaging techniques may effectively discern tumor boundaries. An albumin-based nanoprober has been developed along with catalase biomimetics to enhance the efficacy of phototherapy and imaging modalities. The photo theragnostic nanoprober could facilitate the activation of fluorescence and photoacoustic imaging, along with infrared thermal imaging, to allow for the distinction between normal cells and malignancies [130].

## 10. Imaging Based on Nanoengineering of Mesenchymal Stem Cells (MSCs)

Nowadays, stem cell therapy has emerged as an innovative approach for the diagnosis and treatment of diseases [131,132], especially for brain tumors [133,134]. MSCs have been isolated from different sources including bone marrow [131], adipose tissue [132], dental pulp [135], and endometrium [136], and various methods by intracranial, intrathecal, intravenous, and intratumoral routes have been utilized to deliver these cells and to visualize and diagnose the abnormalities in brain tissue. Administering contrast agents through the carotid arteries is advantageous due to minimal build-ups outside the targeted area to reduce the toxicity and enable the efficient penetration of the brain. In MSC transplantation, MRI and PA imaging as the most common methods can precisely and quantitatively assess the presence of any tumor such as GBM by generating the signals [137]. MSCs labeled with multifunctional gold-coated SPIONs for gold-tagged and unlabeled modalities have been compared by MRI and PA imaging techniques after injection into the internal carotid artery. Gold-tagged MSCs labeled with SPIONs were observed at the tumor site without any harm to the cells or any change in their characteristics. They retained their ability to carry cargo even 72 h after being injected [138].

When the intraparenchymal injection technique was employed, the nanoparticles of bicyclononyne (BCN) (6.10.0) were shown to be linked with chitosan glycol and to be loaded with oleic acid-coated SPIONs and a near-infrared fluorescent dye. The nanoparticles successfully improved the imaging sensitivity of human MSCs in a mouse model and exhibited an exceptional efficacy in vitro and in vivo [139]. Additionally, a nanogel containing minuscule iron oxide nanoparticles was developed to interact with stem cells for the MRI of brain cancers. The intravenous injection of the nanogel led to a notable improvement in MR signals and accumulation, specifically, at the tumor area, and it surpassed the cell-free nanogel performance [139].

A nanocomposite customized for glioblastoma by utilizing MSC-based technology termed LPLNP-PPT-TRAIL with long-term stability was developed. It was shown that near-infrared luminescence could successfully track the movement of modified MSCs in glioblastoma, revealing these nanocarriers to be a specific diagnostic tool [140]. As MSCs can penetrate the BBB, they can be used as an effective vector for targeted cellular imaging. So, paired MSCs were employed to track the precise location of these nanocarriers within pseudo-glioblastoma stem cells. SPIONs, when coated with PEG and tagged with MSCs after intravenous injection, can enhance their movement towards glioblastoma and can be detected in glioblastoma by an in vivo MRI [141].

## 11. Discussion

High-grade gliomas are considered the most common malignant and aggressive neoplasms in the CNS. They are primary brain tumors arising from neuroglial progenitor cells that have been histologically categorized into astrocytic, oligodendroglial, and ependymal tumors. These tumors have a therapeutic challenge due to their tendency to infiltrate and disseminate into surrounding tissues and due to the protective mechanisms in the brain, notably the BBB. These barriers, while crucial for protecting the brain from harmful substances, can significantly hinder the effective delivery of therapeutic drugs to the tissue [142]. Nanotechnology, with its focus on developing materials and devices under 100 nm, has revolutionized various industries, especially those in relation to medicine. Its relevance to medicine is underscored by the fact that nanoparticles are comparable in size to cellular components. This similarity facilitates a wide range of medical applications ranging from drug delivery and biomarker discovery to the modulation of cellular activities [143].

The spectrum of explored nanomaterials is diverse, encompassing liposomes, dendrimers, CNTs, nano-micelles, polymersomes, gold nanoparticles, nanogels, QDs, and magnetic nanoparticles [2,5,9]. These nanomaterials have a high surface-to-mass ratio, which is a critical factor in their ability to bind, absorb, or carry other molecules. Recent advances in nanotechnology have led to the development of nanoparticles as potential diagnostic and therapeutic tools in the field of neurological cancers and neuro-onclogy [143,144]. Using 186-Rhenium liposomes for GBM has resulted in heightened radiation doses that target the tumor. This highlights the potential of nanoparticles as a targeted delivery system for cancer therapies that enhances therapeutic impact and minimizes adverse effects [143,145].

Lipid-based nanoparticles (LBNPs) are a category of compounds that have been utilized to treat various diseases, most notably malignancies. Liposomes are now the most often utilized LBNPs because of their excellent biocompatibility and flexibility, although solid lipid nanoparticles (SLNs) as well as nanostructured lipid carrier (NLCs) have gained popularity lately. LBNPs, as a class of nanoparticles, are important in brain cancer therapy [146]. But, besides their diversity, liposomes are widely employed based on their great biocompatibility and ability to encapsulate a wide range of cargos. A few of LBNPs (for example, Doxil or Abraxane) have previously received license for brain cancer therapy [147,148], revealing the important role of LBNPs in the treatment of different kinds of brain cancers [149]. Despite their protection and effectiveness, SLNs have many major disadvantages, including their high moisture concentration (70–99.9%), poor drug content due to their crystalline form, drug ejection during preservation, and the potential for polymorphism transitions and development of particles during storage. As a result, changes in the organization of SLNs seem necessary to overcome these constraints. An ongoing research led to the development of a “second generation” of LBNPs at the millennium’s turn named the NLCs with numerous possible applications, while SLNs are at the leading edge of innovations in nanotechnology as well as commercial launch [150].

Gogoi et al., in the co-encapsulation of dextran, La0.75Sr0.25MnO3, and iron oxide via encapsulating paclitaxel (PTX) in the hybrid liposome could combine a self-controlled hyperthermia and chemotherapy. In vitro and in vivo studies demonstrated the biocompatibility and therapeutic efficacy of this new magnetic liposome [150]. Lakkadwala et al. designed a modified liposome in the treatment of brain cancer by penetrating peptides and transferrin to the encapsulate 5-fluorouracil (5-FU). They showed that this dual-functionalized liposome could transport the anticancer drug through the BBB and deliver 5-FU to tumor cells [44]. Liposomes, with lots of advantages, also have versatile and adaptable nanocarriers for drug delivery to the brain. They, through the encapsulation of hydrophilic and hydrophobic drugs, biocompatibility, and biodegradability, can accumulate in tumor cells, improve the stability of chemotherapeutic agents, decline the toxicity, and, finally, increase efficacy [34]. Liposomes, through the incorporation of site-specific ligands, can enable the active targeting of tissues, lead to the release of localized drugs, and minimize the adverse and side effects on healthy tissues and organs. Also, due to the resemblance of liposomes to cell membranes, they can efficiently permeate biological membranes, enhancing the efficacy and therapeutic advantages of the drug [34].

There are also some limitations for liposomes, as the BBB poses a significant challenge for drug delivery to the brain. Nowadays, liposomal formulations have been designed to overcome this limitation and have demonstrated effectiveness in overcoming this challenge via lots of BBB-facilitated transport mechanisms. Still, many obstacles and challenges impede the translation of liposomal-based therapies in brain tumors from the laboratory to clinical practice. With all the progress made in recent years, there are still many unanswered questions. Liposomes have a few drawbacks too that are correlated to the physical and chemical characteristics of these nanoparticles. Physical instability may happen because of changes in the temperature, freezing and thawing cycles, and also due to mechanical stresses of storage or handling processes. Chemical degradation, such as hydrolysis or lipid oxidation, may impact their stability. To reach long-term stability for liposomes, many approaches have been undertaken. Cryopreservation and lyophilization were shown to help minimize degradation through the removal of water content or freezing of liposomes at ultra-low temperatures. The incorporation of stabilizing agents such as surfactants, cholesterol, or polymers can inhibit aggregation and decrease leakage. The selection of lipids with high stability and the incorporation of antioxidants can protect liposomes from degradation. Other challenges and obstacles include biosafety evaluations by health regulatory authorities, human biological issues in relation to liposome administration, commercial-scale production, and their high costs [34].

Currently, verified criteria to evaluate the safety of nanodrugs, including liposomes, are absent. Traditional approaches to assess the safety of conventional drugs may not precisely determine the safety of liposomes. Modifications in physicochemical characteristics including surface area, shape, size, and aggregation at larger scales can influence biodistribution and interactions with cells and biomolecules that can later affect the safety investigations. Also, alterations in reagents, synthetic routes, routes of administration, or manufacturing processes can impact the toxicity profile and require the re-assessment of the drug’s safety [34]. For liposomes to be utilized in human patients, they are required to be synthesized based on a large scale and with consistent reproducibility. Lots of formulations used for liposomes could not enter the market based on problems in the scaling-up of the production or issues in reproducibility. It is necessary to mention that the complex nature of liposomes adds to the challenging issues of large-scale manufacturing. Liposomes were typically processed in limited quantities in preclinical and clinical trials as it was easier to manage and optimize the formulations on a small scale. Limitations in manufacturing processes in large-scale production were shown to result in significant alterations in physicochemical characteristics, drug content, surface charge, size, and therapeutic outcomes. Also, the increased cost of scaling up in liposomal manufacturing can decrease the widespread application of innovative formulations for liposomes. The absence of standards and rules for methods of liposomal production, safety and efficacy evaluations, and quality control can lead to limitations in their development and clinical translation. So, there is still a lack of accepted regulatory criteria in the utilization of liposomes in clinical practice [34].

Nanotechnology has attracted great attention and interest in the field of medicine and biology. During the last 20 years of its application in biomedicine, the modified CNTs have shown better biocompatibility and a multimodal functionality. CNTs can present several advantages including high drug loading, good penetration, photothermal ablation, and inherent diagnostic capabilities that make them an excellent choice in cancer therapy, which opens an opportunity for “customized medicine”, where diagnostic as well as therapeutic choices can be made based on the molecular characteristics of the patient. Many studies have demonstrated that CNTs can be used in different cancer imaging methods such as PA, Raman, radionuclide, and NIR fluorescence imaging approaches as well as MRI for cancer diagnosis. In addition, when utilized in combination with other diagnostic techniques, CNTs can be used in nanobiosensors for the early detection of various types of cancers with high specificity such as pancreatic, liver, and ovarian cancers. The use of CNTs, which are comparable to the scale of biomolecules, can be an important tool for targeted drug deliveries. The exploration of CNTs as carriers for drug delivery to the CNS has been an important area of scientific inquiry. Although CNTs exhibit low solubility in water in their unaltered form, they can be engineered to cross the robust BBB through careful functionalization and modification and open new possibilities in therapeutic interventions within the CNS [151].

The unique physical and chemical properties of CNTs, such as their excellent electrical conductivity, strength, and surface area, have prompted research into their diagnostic potential in CNS-related diseases such as Alzheimer’s disease and Parkinson’s disease [152]. These properties offer unique opportunities for advances in the detection and monitoring of such neurological disorders [153]. CNTs allow for the more accurate and earlier detection of biomarkers and disease-induced changes, which may aid diagnostic and therapeutic strategies [154]. The investigations on CNTs for medical applications have not been without drawbacks too, particularly regarding their possible interactions in the CNS. The concerns about the potential toxicity of CNTs have emphasized the need for thorough research and the understanding of their possible effects on neural tissue and function [155]. CNTs can enter the CNS through a variety of pathways, including crossing the BBB and blood–spinal cord barrier via the body’s circulation. In addition, they can enter the brain through the nasal and olfactory routes as well as the trigeminal nerve branches in the olfactory and respiratory regions. Upon interaction with brain cells, CNTs can stimulate microglia and astrocytes to release a range of mediators/chemicals that may lead to inflammation, apoptosis, or oxidative stress in the brain [156].

MWCNTs exhibit similar interactions with neural tissues with a toxicity profile comparable to that of SWCNT. A study conducted on the interaction of multi-walled CNTs with neural tissues showed that the direct injection of multi-walled CNTs into the brains of mice can lead to neuroinflammatory responses and reveal how these nanotubes affect neural health [157]. When MWCNTs are absorbed, they trigger a range of neuroinflammatory responses with a significant increase in inflammatory cytokines in the cortex. There is growing evidence that shows that the surface oxidation of CNTs may play an important role in the replenishment of energy. It results in elevated cytokine levels, glial cell activation, and inflammatory responses within brain cells. So, a thorough examination of the interactions of CNTs with neural tissues is essential to ensure the safety and effectiveness of CNTs in CNS-related diagnostics and therapeutics interventions [158].

The defined standards in the treatment of newly diagnosed cases of GBM are the extensive surgical resection of the affected tissue and a concurrent chemoradiation through temozolomide and chemotherapy by adjuvant temozolomide [159]; this therapeutic approach is aggressive and its survival outcome is still marginal [160]. Therefore, an increasing demand exists to look for novel and efficacious therapeutic regimens for GBM. During the past two decades, exploring ways for targeted agents and immunotherapies to be less toxic in the realm of neuro-oncology, especially tailored for GBM, has been highlighted to increase patient survival [5]. The molecular aspects in relation to GBM enroll mutations in telomerase reverse transcriptase (TERT) promoter, the amplification of the EGFR gene, and variations in chromosome copy numbers (+7/−10) [2]. So, GBM can demonstrate histological and molecular heterogeneities that can lead to a significant resistance to therapeutic regimens [6]. Genetic mutations and chromosomal changes such as loss of chromosomes 10 and 9p and gaining chromosomes 7 and 19 exhibit the potential need for gene-oriented therapies in GBM [6,8], which has attracted significant interest in the last decade to several medical approaches such as CRISPR-Cas9 system in gene editing technology [5]. CRISPR has been introduced as the fastest, cheapest, most versatile, and most reliable and available gene editing tool to uncover genetic alterations, oncogenic targets, and epigenetic regulation. CRISPR-Cas9 has been considered as the preferred choice for gene editing or the editing of genomes in various cancers, including GBM [5,19,20,21]. Considering the current trend in medical research towards more accessible treatments for diverse pathologies to possess broader treatment applicability in low- and middle-income countries, gene editing has attracted huge interest, especially in cases of GBM [5,19,20,21].

In this review, we discussed nanoparticle-based approaches in the diagnosis and treatment of brain tumors. However, there may be still several limitations in the clinical use of nanoparticles. To design nanoparticles for clinical application, it is necessary to recognize the transport regulation mechanism of BBB materials and also optimizing the nanoparticles. An accurate investigation into the mechanism and factors in relation to nanoparticles affecting the brain and improving synthesis technology is important to explore new and promising delivery systems for a clinical setting. Also, consistent criteria and decision modalities for the in vivo biodegradability of nanomaterials are an essential item on the agenda. It is necessary to mention that adherence to the concept of “all in one” single imaging or treatment functionalities may face the risk of being eliminated in nanotheranostics. Several precision theranostic nanoparticles have been considered an important direction. However, challenges such as biocompatibility and compatibility with the biological environment still persist and require further investigation and scientific innovation. More deep investigations in the mentioned fields and the establishment of a mature nano-diagnosis and treatment system to be utilized for clinical therapy are just around the corner.

## 12. Conclusions

Recent advances in nanotechnology have opened new avenues for diagnostic imaging and therapeutic interventions to facilitate precise drug delivery to specific tumor regions and to limit damages to healthy brain neighboring tissues when nanoparticle-based approaches are undertaken in the diagnosis and treatment of brain tumors. However, challenges such as compatibility with the biological environment still persist that need further in-depth studies to overcome obstacles in the diagnosis and treatment of brain tumors. So, this review emphasizes the potential efficacy of nanoparticle use for the diagnostic imaging of brain tumors to improve their identification and also in the treatment of brain tumors.

## Figures and Tables

**Figure 1 jcm-13-07449-f001:**
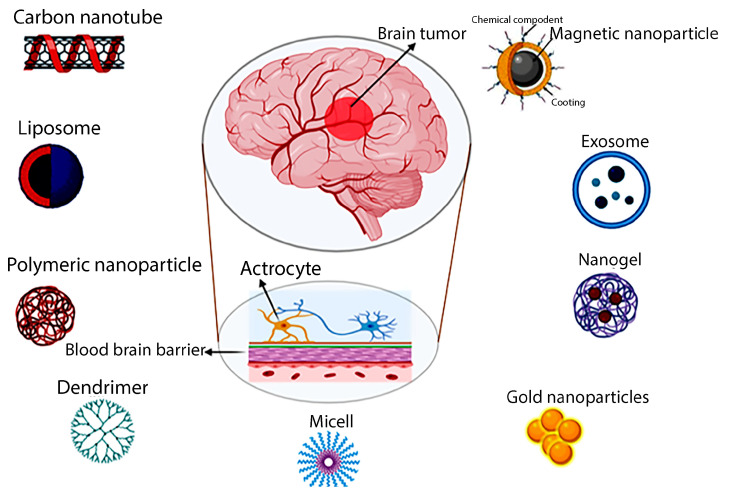
Novel nanoparticles employed in the treatment of brain cancer.

**Figure 2 jcm-13-07449-f002:**
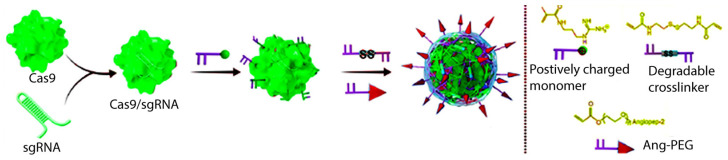
In situ synthesis of disulfide cross-linked nanocapsules containing Cas9/sgRNA and functionalized with angiopep-2 targeting ligand. PEG: polyethylene glycol.

**Figure 3 jcm-13-07449-f003:**
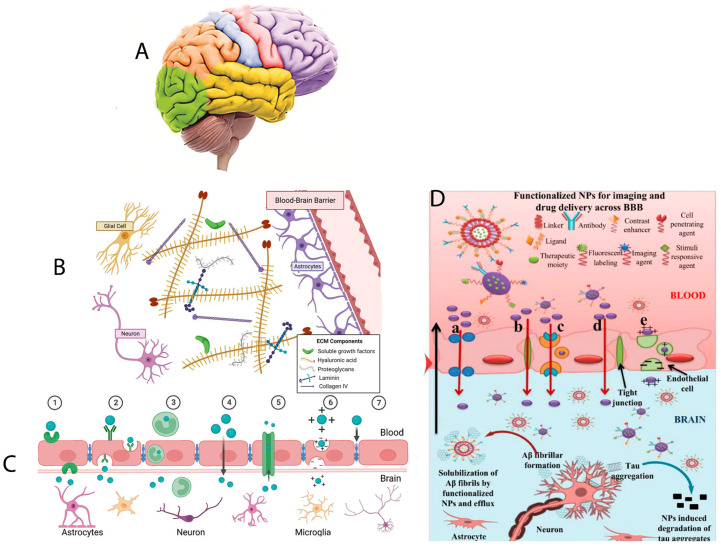
The character, function, and pathways of nanoparticles (NPs) to overcome the blood–brain barrier (BBB) for an efficient delivery of therapeutic moieties in the treatment of brain disorders. (**A**) Image of the human brain. (**B**) Extracellular matrix (ECM) of the brain with a unique composition, including hyaluronic acid, collagen IV, and other components along with glial cells, neurons, and astrocytes. (**C**) Strategies and materials for BBB regulation and brain-targeted drug delivery: a schematic diagram of different mechanisms for BBB crossing. (**D**) Functionalized NPs for imaging and targeted drug delivery to the brain with Alzheimer’s disease (AD), while different pathways of transport (**a**–**e**) across the BBB are utilized by functionalized NPs. (**a**) Transport of NPs through cellular transport proteins. (**b**) Transport of NPs through tight junctions. (**c**) Transport of NPs via receptor-mediated transcytosis. (**d**) Transport of NPs via the transcellular pathway following diffusion, specifically adopted by gold NPs [36,37,38].

**Figure 4 jcm-13-07449-f004:**
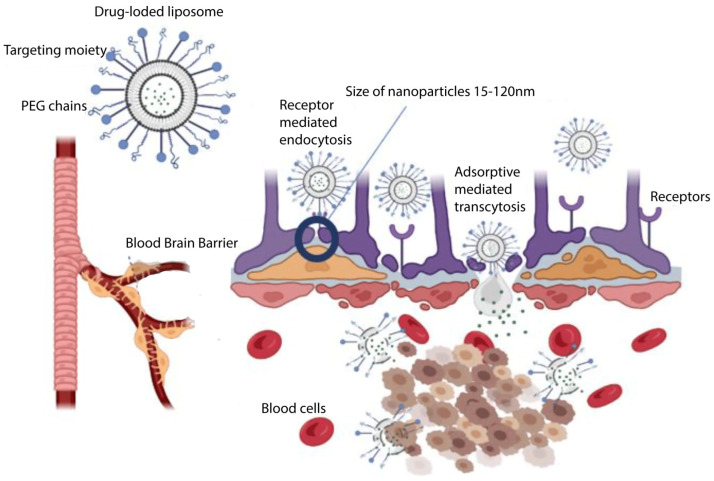
Schematic illustration of liposomal drug delivery across the blood–brain barrier. PEG: polyethylene glycol [34].

**Table 1 jcm-13-07449-t001:** In vivo preclinical investigations on liposomal drug delivery to the brain tissue [36].

Mechanism of Entry	Type of Liposome	Drug Payload Size	Liposome Encapsulation	Efficacy	Summary	Reference
Not mentioned	Liposomal Dox	Dox	Not mentioned	Not mentioned	Evaluation of treatment with FUS + Dox in 9L of rat glioma resulted in substantial improvement in survival and tumor shrinkage. Dox alone showed minimal effects.	[44]
Not mentioned	Liposomal TMZ	TMZ	148.13 ± 2.66 nm	53%	TMZ-lipo together with ultrasound irradiation showed an increase in tumor-cell-killing efficiency compared to TMZ alone.	[45]
Receptor-mediated endocytosis	PEGylated liposomes	Dox and CB	212 ± 10 nm	83.9%	Enhanced survival rate in brain cancer cells using PEG-lipo nanoparticles.	[46]
Receptor-mediated endocytosis	Liposomes	CPPs and Tf	155 nm	87.4 ± 3.85%	Higher brain tumor accumulation was shown in comparison to the control.	[47]
Receptor-mediated endocytosis	RI7217 (mouse transferrin) and muscone-conjugated liposomes	DTX	159.1 ± 4.4 nm	65.37 ± 0.87%	Improved survival and tumor absorption in U87-MG cells.	[48]
Receptor-mediated endocytosis	RGD-TPCS-theranostic liposomes	DTX and QDs	157.6 ± 3.2 nm	68.41 ± 3.56%	RGD-TPCS-theranostic liposomes enhanced survival and tumor targeting efficacy.	[49]
TML	TML	CPT-11 and Dox coated with magnetic Fe_3_O_4_ nanoparticles and conjugated with CET	193.7 ± 2.3 nm	87.9 ± 1.4%	Significant tumor shrinkage and high biocompatibility via magnetic guidance.	[50]

CB: Carboplatin. CET: Cetuximab. CPPs: Cell-penetrating peptides. CPT-11: Camptosar-11. DTX: Docetaxel. Dox: Doxorubicin. FUS: Focused ultrasound. PEG: polyethylene glycols. QDs: Quantum dots. RGD: Arginylglycylaspartic acid. Tf: Transferrin. TML: Thermosensitive magnetic liposomes. TMZ: Temozolomide.

**Table 2 jcm-13-07449-t002:** Comparison of nanoparticle-based strategies for precise diagnosis and therapeutic intervention of brain tumors [43].

Imaging Methodology	Criteria for Selection	Utilized Nanostructures	References
MRI	The imaging technique is very responsive to alterations in the cartilage and bone. It offers a detailed visualization of the anatomical composition of the brain and highlights the differences in soft tissues	The surface of iron oxide nanoparticles is adorned with peptides	[78,79]
SERRS	Having high specificity that provides data about the location of biochemical components of cells	SERRS NPs with 68Ga consisted of gold core and silica shell	[80,81]
PET	The nuclear imaging approach used to detect and analyze pathophysiological alterations in the brain, with the advantage of being able to penetrate deeply into the tissue	The nano-system consisted of amphiphilic dendrimer self-assembled with gado-fullerene NPs that have been modified with alanine	[82,83]
CT is a medical imaging technique that uses X-rays and computer processing to create detailed cross-sectional images of the body	Capable of discerning variations in electron density between tissues for the purpose of establishing a diagnosis	Liposomes that are linked with transferrin and lanthanide nanoparticles	[84,85]
Utilizing multiple imaging techniques simultaneously	The ability to correlate with cell density in order to comprehend the variability of the tissues and exhibit a high level of sensitivity and specificity	A gold nanostar with a silica covering, functionalized with PEG, and embedded with a gold core, used as SERRS-MSOT nanoprobes. These nanoprobes have also SERRS-MRI capabilities	[86,87]
PA is a technique that combines light and sound to create images	Obtains molecular data with exceptional precision in real-time and can be employed concurrently with other imaging methodologies	Silicon quantum sheets and molybdenum di-sulfide nanosheets that are linked with indocyanine green	[88,89]
FL is a technique used to visualize and study the distribution of fluorescent molecules or markers in a sample	Non-invasive, but with limited spatial resolution	Gold NPs	[90,91]
Ultrasound technology that is concentrated and directed towards a certain target	Immediate visualization of brain structure using three-dimensional pictures enhanced with contrast	Cisplatin-gold NPs conjugated with mesoporous organo-silica NPs	[92,93]

CT: Computed tomography. FL: Fluorescence imaging. MRI: Magnetic resonance imaging. MSOT: Multispectral optoacoustic tomography. NPs: Nanoparticles. PA: Photoacoustic imaging. PEG: Polyethylene glycol. PET: Positron emission tomography. SERRS: Surface-enhanced resonance Raman scattering imaging.

**Table 3 jcm-13-07449-t003:** Multimodal imaging characteristics with dendrimer-type nanomaterial [115].

Dendrimer Type	Contrast Agent	Conjugate	Targeting Ligand	Imaging Modality	Cell Line/Animal Model	References
Phenylalanine dendrimers	99mTc	99mTc-labeled dendrimer-phenylalanine conjugate	-	SPECT	C6 Glioma cells	[116]
G5 PAMAM	131I	Radionuclide	MMP-2 overexpressing tumor cells	SPECT	Mouse	[117]
G3 poly-L-lysine dendrigraft	DTPA-Gd	Gd-DTPA-D3-PEG-CTX	Chlorotoxin for MMP-2 receptors	MRI	Mouse	[118]
G3 poly-L-lysine dendrigraft	DTPA-Gd	-	-	MRI/PET/SPECT	C6 Glioma cells	[119]
G2 PAMAM	Macrocyclic Mn (II)	RGD-Au-Mn DENP	RGD peptide for αvβ3 integrin	CT and MRI	Mouse	[120]
G3 poly-L-lysine dendrigraft	DTPA-Gd	SP-PEG-DGL-DTPA-DAEIP	Endogenous neuropeptide-binding NK-1 receptor	MRI	Mouse	[121]
G5 PAMAM	SPIONs	5NHA-RGDFe3O4 NPs	RGD peptide for tumor cells expressing αvβ3 integrin	MRI	Mouse	[122]
G5 PAMAM	DTPA-Gd	GdDTPA-PAMAM-PEG-T7	T7 peptide for cells expressing transferrin receptors	MRI	Mouse	[123]
G5 PAMAM	131I	131I-labeled modified dendrimer with peptide	Dendrimer nanoplatform, radionuclide cancer therapy	SPECT	Mouse	[124]
Polyglycerol dendrimers	Boron-dipyrromethene	Boron-dipyrromethene-conjugated polyglycerol dendrimers	Fluorescent probe	Single-molecule optical imaging	-	[125]

CT: Computerized tomography. DENP: Diethyl P-nitrophenyl phosphate. DTPA-Gd: 1,2-distearoyl-sn-glycero-3-phosphoethanolamine-N-diethylenetriaminepentaacetic acid (Gadolinium salt). MMP-2: Matrix metalloproteinase-2. MRI: Magnetic resonance imaging. NPs: Nanoparticles. PAMAM: poly(amidoamine). PEG: Polyethylene glycol. PET: Positron emission tomography. RGD: Arginine-Glycine-Aspartate. SPECT: Single photon emission computed tomography. SPIONs: Superparamagnetic iron oxide nanoparticles.

## Data Availability

The data analyzed in this study are available from the corresponding author based on reasonable request.

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
