# Peer review of "Nanoparticle-Based Approaches in the Diagnosis and Treatment of Brain Tumors"

_jcm, 2024, doi:10.3390/jcm13237449_

Round 1
Reviewer 1 Report
Comments and Suggestions for Authors
Brief summary: This article is a literature review about the most important nanoparticles being studied for glioblastoma treatment. They discuss the multiple uses of the nanoparticles, including as drug carriers and diagnostic agents. Learning about the new technologies being developed to overcome the multiple barriers malignant and aggressive cancers possess is of the utmost importance. Discussing these topics with the medical community may inspire many scientific leaders to evaluate the pharmacodynamics of the nanoparticles and conquer the challenges these could have.
Specific comments:
-
Page 2. Introduction: “Brain tumors arise from different types of cells within the central nervous system (CNS)...”. It is important for the authors to specify early in the paragraph (and the manuscript) if they are referring to primary or secondary brain tumors, i.e. metastasis.
-
Page 3. Introduction: “So, drug delivery systems based on nanoparticles were demonstrated to solve this problem and provide the possibility of encapsulation, protection, and transport of therapeutic agents through the brain towards the tumor cells.” Please, attach reference.
-
Page 5. Introduction: “Also, the drug-loading efficiency is super high, almost 100%...” Change the adjective “super” for “very”, for a more professional tone
-
Page 6. Introduction: “Lu et al. utilized knockdown of BIG1 and BIG2 to significantly decrease VEGF mRNA and protein levels in GBM U251 cells…” This is the first time the authors use VEGF; must explain the abbreviation the first time used.
-
Page 22. Nano-micelles: “Coating these micelles with hybrid membrane mUMH could enhance…” What does mUMH mean? Not explained previously in the manuscript.
-
Page 26. Extracellular vesicles (EVs) and exosomes: “The body small subset cells release nanoscale particles that are surrounded by a membrane.” If this is an explanation for the extracellular vesicles, it should be stated early in the sentence.
-
Page 31. Multimodal imaging with nanomaterials: “A liposome carrier possessing theranostic characteristics was deliberately engineered to selectively target glioma…” If this is a study done by the authors of the reference [124], it should be stated at the beginning of the sentence.
-
Page 31: Multimodal imaging with nanomaterials: “A further study examined the effects of nanomaterials in brain tumor imaging…” If this study was done by other authors, the sentence should start with something like “In another study, the authors examined the effects…”
-
Page 32. Imaging based on nanoengineering of mesenchymal stem cells: “MSCs based therapies utilize MRI and PA imaging…” The abbreviation “MSCs” is not previously explained; please, explain.
-
Page 33. Discussion: “Brain tumors are considered the most frequent type of cancer…” This detail is incorrect. Brain tumors are not the most common type of cancer, at least in the US. If these statistics are specific for a country or a city, it should be explained.
-
Page 35. Discussion: “Ongoing research led to development of a "2nd generation" of LNPs at the millennium's turn…” If the abbreviation LNPs is the same as LBNPs, it should always be written the same way throughout the article.
Page 35. Discussion: “Furthermore, liposomes enhance the stability of chemotherapeutic drugs, reduce toxicity, and increase efficacy.” Please, attach reference.
Grammar/Syntax comments:
-
Page 2. Abstract: “Among glioblastomas, Glioblastoma multiforme (GBM) is the most aggressive and prevalent malignant brain tumor for them nanomaterials, such as metallic and lipid nanoparticles, and quantum dots have been acknowledged as efficient carriers to traverse the blood-brain barrier, integrate and reach the necessary regions for neurooncology imaging and treatment of brain tumor.” This sentence is extremely long and the readers may get lost reading it. I recommend the authors to divide it in more than one sentence. Transition between the sentences is also important.
-
Page 2. Introduction: “Glioblastoma is one of the most common rare gliomas…” This sentence may sound more clear if the authors just write “Glioblastoma is one of the rarest gliomas…”
-
Page 3. Introduction: “When nanoparticles were targeted with ligands such as antibodies or peptides, the process of delivering drugs to specific locations was facilitated and led to an increased therapeutic efficiency, reduced side effects, and significant improvements in the management of brain tumors.” In this sentence, the verbs should be changed to ‘present’ for improved readability. Writing in past language is not always correct.
-
Page 4. Introduction: “Radiation therapy has also been employed to circumvent brain malignancies by utilizing accurate and concentrated radiation that specifically aimed at the tumor area; while safely preserved the neighboring brain tissues”. Change “preserved” to “preserving”.
-
Page 5. Introduction: “Nanocapsules by employing clustered regularly interspaced short palindromic repeats…” The “by” in this sentence can be removed.
-
Page 5. Introduction: “In addition, the small size of the nanocapsule, around 30 nm, and the rational modification of angiopep-2 peptide, a peptide with good affinity with lipoprotein receptor-related protein-1 (LRP-1), which is highly expressed on BBB endothelial cells and glioblastoma cells allow them to cross the BBB and to efficiently undertake the intracellular delivery to GBM cells.” This sentence is extremely long. Again, authors can divide the sentence or place parentheses for better readability.
-
Page 7: Introduction: “Therefore, there is a need in future researches to improve nanoparticle…” The word “researches” does not exist; plural is research.
-
Page 7. Nanotechnology in brain cancer therapy: “Their favorable physicochemical characteristics, such as their small size, their large surface area relative to volume, their specific structural characteristics and the possibility of attaching different molecules to their surface, transforming into excellent transport vehicles to cross cellular barriers including BBB are the main related factors in development of new diagnostic and treatment approaches of brain cancers in recent decades.” This sentence is too long and should be divided for better understanding. The word “transforming” should be changed to “transform them into…” or “make them into…”
-
Page 8. Nanotechnology in brain cancer therapy: “…even there are exceptions such as certain metal/metal oxide nanoparticles that their biodegradability is dependent on coating and synthesis methods employed.” This can be a second sentence, starting with “However, there are exceptions…”.
-
Page 9. Nanotechnology in brain cancer therapy: “...various strategies were developed for regulation of BBB permeability as well as library of brain-targeted drug delivery systems (Figure 3).” Add an “a”, as follows → “as well as a library…”
-
Page 9. Nanotechnology in brain cancer therapy: “Transport routes of the drug molecules across the BBB occurs via the pathways including…” Write “via many pathways, including…” instead of “via the pathways…”.
-
Page 10. Liposomes: “…while they have an onion- like structure in a multilamellar structure.” This sentence can be changed to "while the multilamellar structure has an onion-like structure" for better readability.
-
Page 10. Liposomes: “The efficiency of liposome encapsulation increased with liposomal size and decreased…” Write verbs in present: increases, decreases.
-
Page 11. Liposomes: “Liposomes have the potential to enhance drug delivery to brain when a brain tumor exists and can improve the treatment efficacy.” Change sentece to “...enhance drug delivery to the brain when a tumor exist…”, and delete the second word “brain”
-
Page 17: Liposomes: “Liposomes loaded with erlotinib and doxorubicin showed promise effect…” Correct word is “promising”, not “promise”.
-
Page 17. Liposomes: “These liposomes when modified with transferrin demonstrated…” Add commas as follows → “These liposomes, when modified with transferrin, demonstrated …”
-
Page 17. Liposomes: “...in mice model…” Change to “models”, in plural.
-
Page 17. Liposomes: “So these liposomes have been promising drug delivery systems…” Change to “These liposomes are a promising drug delivery system…”
-
Page 17. Liposomes: “Immunoliposomes which are liposomes modified with antibodies have also held significant promise in cancer therapy such as Food and Drug Administration (FDA)-approved monoclonal antibodies (mAbs) that have been designated for cancer treatment.” This sentence can be divided in two for better readability.
-
Page 18. Liposomes: “Paclitaxel with the efficacy in targeting microtubules and combating a range of cancers such as lung, ovarian, and brain tumors was demonstrated to have…”. This sentence requires commas to divide the explanation of Paclitaxel. The sentence can be changed as follows: “Paclitaxel, with its efficacy targeting microtubules and combating a range of cancers such as lung, ovarian, and brain tumors, has demonstrated to have…”
-
Page 18. Liposomes: “...because of the low levels of paclitaxel that reach the gliomas and poor penetration to the BBB [54, 55].” Change to “penetration of the BBB”.
-
Page 18. Liposomes: “The sphere size can affect the drug release, while the smaller spheres have a faster release…” Change to: "...with the smaller spheres having a faster release"
-
Page 19. Dendrimers: “Dendrimers as hyper-branched nano-sized polymers…” Add commas to separate the explanation. → “Dendrimers, as hyper-branched nano-sized polymers, are recognized…”
-
Page 19. Dendrimers: “Therefore, targeted ligands can be added to dendrimer surfaces for selective delivery to brain tumor cells and sparing normal cells” Delete the “and” before “sparing normal cells”
-
Page 19. Dendrimers: “Dendritic polymers, specifically PAMAM dendrimers have become highly significant in the field of cancer therapy because to their…” Change “because to their” to “because of their…”
-
Page 20. Dendrimers: “Bio-distribution study revealed…” Start the sentence with an “A…” or a “The…”
-
Page 20. Dendrimers: “Overall, the study suggested that PCT by utilizing chitosan-anchored dendrimers for TMZ delivery could hold promise for enhancing (280)glioblastoma therapy [60].” This sentence could be rewritten as follows for clear readability: “Overall, the study suggested that utilizing chitosan-anchored dendrimers for TMZ delivery could hold promise for enhancing glioblastoma therapy”
-
Page 20. Dendrimers: “This is due to the high-branched…” The correct adjective is “highly branched”
-
Page 20. Dendrimers: “...available internal cavities of these polymers which make them excellent delivery systems for genes and drugs [54].” Add a comma after “polymers”, before “which”.
-
Page 20. Nano-micelles: “There are several nanomedicines such as nanoparticles, niosomes, liposomes, dendrimers, micelles, etc…” The enumerations can be placed within a parenthesis.
-
Page 20. Nano-micelles: “Among all nanomedicines, micelles are one of the best suitable nanocarriers which facilitate higher penetration of therapeutic drugs to BBB, reduce multidrug-resistance and inhibit tumor recurrence post-surgery [61].” This can be divided as follows → “ Among all nanomedicines, micelles are one of the best suitable nanocarriers. They facilitate higher penetration of therapeutic drugs to BBB, reduce multidrug-resistance and inhibit tumor recurrence post-surgery [61].”
-
Page 21. Nano-micelles: “Micelles are absorbed through passive diffusion and receptor-based endocytosis mechanism has clearly cited…” This sentence can be divided as follows: “Micelles are absorbed through passive diffusion and the receptor-based endocytosis mechanism. This has clearly demonstrated…”
-
Page 21. Nano-micelles: “They can boast unique properties such as stability, improved biological compatibility, prolonged plasma circulation and enhance penetration…” Correct word is “enhanced”
-
Page 21. Nano-micelles: “HK-Dox-MPEG-PCL micelles could effectively suppress proliferation of glioma cell…” Write “cells”, in plural.
-
Page 24. Silver nanoparticles (AgNPs) and gold nanoparticles (AuNPs): “Although there were noticeable variations in the rates of cell death, levels of active caspase 9 and active caspase 3 between the control group and the group treated with AgNPs, the inhibitory effects of AgNPs on cell growth were more prominent than their impact on cell death [70].” I don’t understand this sentence very well. It should be rewritten for clarity.
-
Page 24. Nanomaterials for brain cancer diagnosis and biosensing: “Novel methods by controlling the targeting of nanostructures and by utilizing specific coatings were demonstrated that can precisely evaluate tumor tissues [30].” Sentence should be rephrased as follows: “...were demonstrated to precisely mark/pinpoint tumor tissue..."
-
Page 26. Extracellular vesicles (EVs) and exosomes: “In individuals with glioma, EVs containing EGFR proteins that can be found in the serum…” The word “that” should be deleted.
-
Page 27. Multimodal imaging with nanomaterials: “Meanwhile, gadolinium (Gd)-chelated diagnostic agents were extensively utilized as contrast agents for tumor imaging [109].” Change “were” for “are”; we are currently using gadolinium for tumor imaging.
-
Page 28. Multimodal imaging with nanomaterials: “Recently, Rasouli et al. investigated the potential of 99 mTc-labelled dendrimer-phenylalanine conjugates in C6 glioma cell lines for diagnosis of brain tumors using single photon emission computed tomography (SPECT) and showed that these dendrimers did not exhibit toxicity in the brain, whereas phenylalanine increased the accumulation and deposition of 99 mTc-labelled dendrimer in brain tumors [113] (Table 3).” Very long sentence. Can be divided for clarity and improved readability. The word “labelled” is incorrect; the correct word is “labeled”.
-
Page 33. Imaging based on nanoengineering of mesenchymal stem cells: “...while near-infrared luminescence could successfully track the movement of modified MSCs in glioblastoma revealing that these nanocarriers can be used as a specific diagnostic tool [136].” This sentence can be divided from the previous and start with “Near-infrared luminescence could successfully…”
-
Page 33. Imaging based on nanoengineering of mesenchymal stem cells: “SPIONs when coated with PEG were tagged with MSCs to enhance their movement towards glioblastoma after intravenous injection that could be detected in glioblastoma by an in vivo MRI [137].” This sentence has very poor clarity and I do not understand the message the authors are trying to deliver.
-
Page 35. Discussion: ”...previously received licensed for brain cancer therapy…” It should be written “...previously received a license for brain cancer therapy…”
-
Page 35. Discussion: “SLNs beside their protection and effectiveness, they have many major disadvantages…” A better wording may be “Despite their protection and effectiveness, SLNs have many major disadvantages…”
-
Page 36. Discussion: “The BBB poses a significant challenge to drug delivery to the brain…” Sentence should say “The BBB poses a significant challenge for drug delivery to the brain…”
-
Page 37. Discussion: “Nanotechnology has attracted great attention and interest in the field of medicine and biology, and during the 20 years of its application in biomedicine, progressively modified CNTs have possessed better biocompatibility and multimodal functionality.” This sentence should be divided for improved clarity and readability.
-
Page 39. Discussion: “There is growing evidences that shows CNTs surface oxidation may play an important role in the replenishment of energy, focusing on elevated cytokine levels, glial cell activation and inflammatory responses within brain cells emphasizing that a thorough examination of the interactions of CNTs with neural tissues is essential to ensure the safe and effective use of CNTs in CNS-related diagnostics and therapeutics [154].” The correct word is “evidence”, without an “s”. This sentence is very long and poorly understandable. Please, rephrase.
Page 41. Discussion: “This review emphasizes the potential and efficacy of nanoparticle use for diagnostic imaging…” This sentence should say “This review emphasizes the potential efficacy of nanoparticle use for diagnostic imaging…”, unless the authors are using the word “potential” as a synonym for “ability”. In that case. They should rephrase the sentence.
Reviewer 2 Report
Comments and Suggestions for Authors
Abstract:
The abstract must be structured and developed (it must point out exactly the stages that lead to the conclusion).
The terms used must be according to the latest WHO (the same aspect throughout the manuscript).
Emphasize the purpose of the study and its fulfillment, as well as the conclusions.
Introduction:
Tumors can develop either locally or spread to other parts of the body, resulting in the production of non-cancerous or cancerous growths [2, 3]. - can be removed
astrocytoma, anaplastic astrocytoma and glioblastoma - update the information
After "with brain tumors still remains dismal and necessitates innovative strategies to improve outcomes" - continue with a few sentences about prognostic factors that influence survival (DOI: 10.3390/clinpract12050073 and similar studies), then add the purpose of the study "The main goal of this review is to provide…”
In recent years, nanoparticles have gained attention in the management of brain tumors - new subchapter
Define GAP
2. Nanotechnology in brain cancer therapy
"Nanoparticle DDSs involve the utilization of nanocarriers comprised of non-toxic monomers" - new paragraph
"Due to recent developments in materials sciences and nanotechnology" - new paragraph
I would recommend that subchapters 3, 4, 5 be passed, like liposomes and dendrimers
3. Nano-Micelles
"In recent years, micelles have emerged as" - new paragraph
9. Multimodal imaging with nanomaterials
99 mTc-, 99m Technetium
11. Discussion
Doxil and Abraxan without registered trademark symbol
2nd - to write second
Add a paragraph with the limitations of the study, as well as the bias of the bibliographic source choices.
12. Conclusion
Develop conclusions based on the pros and cons identified in this narrative review. Also, there must be correspondence between purpose and conclusion.
In this manuscript, are you referring only to IDH wildtype glioblastoma or according to the old nomenclature also grade 4 IDH mutant astrocytoma? Mention this aspect clearly.
If the images have been taken, their source must be added.
Tables - write the summary of the table before the table, and the abbreviations as a footnote to the table.
Avoid the expressions Han and colleagues, preferably Han et al. . Review this aspect throughout the manuscript.
When resubmitting, try to enter the text in the template provided on the journal's website.
Add author contributions according to author instructions.
References must be entered in the MDPI style. Beware of self-citations!
Some statements are too long, reducing coherence.
The plagiarism coefficient is very high, try to revise this aspect through various methods, so that the information remains correct. Some paragraphs are just copied, not summarized.
Comments on the Quality of English LanguageSome statements are too long, reducing coherence.
Round 2
Reviewer 1 Report
Comments and Suggestions for Authors
First of all, I'd like to congratulate the authors for significantly improving their manuscript! They incorporated almost all the comments and recommendations we gave, and the manuscript sounds much more clear and well written. However, there still are a few grammatical errors that I'll add to this e-mail. After they make these corrections, from my standpoint, the manuscript may be published. Thank you for the honor of reviewing such an interesting paper.
Comments of errors found on revision: P6, Introduction: "They are highly expressed on BBB endothelial cells and glioblas-toma cells that allow them to cross the BBB and to efficiently undertake the intracellular delivery to GBM cells." Authors need to specify the "they" they are referring to. Is the LRP-1 or the Angiopep-2 peptide which is highly expressed on BBB? Is confusing. P10, Introduction: "However, certain metal/metal oxide nanoparticles may be considered to be dependent on coating and synthesis techniques." This sentence can be rephrased as "However, there are exceptions to biodegradability, like certain metal/metal oxide nanoparticles, whose biodegradability is dependent on coating and the synthesis methods employed." P13, Liposomes: "Liposomes are either unilamellar (As small as 100 nm and as large as 200-800 nm) or multilamellar (500-5000 nm; consisted of many lipid bilayers with the same center) [32]." Correct verb is "consisting", not "consisted". P15, Liposomes: "As the ca-rotid artery is considered a major blood vessel that provide brain’s blood supply, the in-tracarotid injection method can lead to the direct injection of the drug into the carotid artery; the intranasal injection route also can bypasses the BBB and use nose to deliver the drug to the brain tissue; the intracranial injection approach can be an effective way in di-rect delivery of drugs to specific regions of the brain tissue and the intraperitoneal injec-tion method van provide delivery of the drug via the peritoneal route." Correct word is "bypass", not "bypasses". Also, the authors wrote "van", instead of "can". P15, Liposomes: "Finally, CED can result to establishment of a pressure gradient on the tip of an infusion catheter implanted in the brain tissue in order to administer drugs directly into the interstitial spaces of the brain tissue [34]." This sentence can be rephrased as "The CED can also establish a pressure gradient on the tip of the infusion catheter implanted in the brain tissue, administering drugs directly into the interstitial spaces of the brain tissue [34]. P16, Liposomes: "Food and Drug Administration (FDA) has approved the monoclonal antibodies..." This sentence should start with "The Food and Drug Administration..." P26, Carbon nanotubes: "CNTs are cylindrical nanostructures with promising applications that continues to attracted great attention of pharmaceutical researchers..." Correct wording is "... continues to attract attention from pharmaceutical researchers..." P36 multimodal imaging with nanomaterials: "They showed these dendrimers not to exhibit any toxicity in the brain..." Correct wording is "...these dendrimers do not exhibit any toxicity in the brain..." P42, Discussion: "Gliomas are considered as the most common malignant and aggressive neoplasms in CNS that have therapeutic challenge due to their tendency to infiltrate and disseminate to surrounding tissues and because of the protective mechanisms in brain, notably the BBB and the blood cerebrospinal fluid (CSF) barrier that restrict the use of surgery as a feasible treatment strategy. They are primary brain tumors arisen from neuroglial progenitor cells, that have been histologically categorized into astrocytic, oligodendroglial, and ependymal." For improved readability, this sentence can be changed and divided as follows: "High-grade gliomas are considered as the most common malignant and aggressive neoplasms in CNS. They are primary brain tumors arising from neuroglial progenitor cells, that have been histologically categorized into astrocytic, oligodendroglial, and ependymal. These tumors have a therapeutic challenge due to their tendency to infiltrate and disseminate to surrounding tissues and due to the protective mechanisms in brain, notably the BBB." P46, Discussion: "With all the progress made in recent years, still there are many questions remain to be answered..." Last half of the sentence should say: " With all the progress made in recent years, there are still many unanswered questions ..." P46, Discussion: "Physical and chemical of these nanoparticles..." Physical and chemical what? Properties, characteristics? Need to add the noun. P54, Conclusion: "However, challenges such as biocompatibility and compatibility with the biological environment still persist..." I think the authors are repeating the same statement and it is redundant. They should delete either "biocompatibility" or "compatibility with the biological environment...".Author Response
Please see the attachment.

Reviewer 2 Report
Comments and Suggestions for Authors
Congratulations!
Author Response
Thank you very much.